# The Role of CD38 in the Pathogenesis of Cardiorenal Metabolic Disease and Aging, an Approach from Basic Research

**DOI:** 10.3390/cells12040595

**Published:** 2023-02-12

**Authors:** Munehiro Kitada, Shin-ichi Araki, Daisuke Koya

**Affiliations:** 1Department of Internal Medicine, Hamada Neurosurgery and Internal Medicine Clinic, Wakayama 641-8509, Japan; 2Department of Nephrology, Wakayama Medical University, Wakayama 641-8509, Japan; 3Omi Medical Center, Kusatsu, Shiga 525-8585, Japan; 4Division of Anticipatory Molecular Food Science and Technology, Medical Research Institute, Kanazawa Medical University, Uchinada, Ishikawa 920-0293, Japan

**Keywords:** CD38, NAD^+^, aging, cardiovascular disease, kidney disease, metabolic disease

## Abstract

Aging is a major risk factor for the leading causes of mortality, and the incidence of age-related diseases including cardiovascular disease, kidney disease and metabolic disease increases with age. NAD^+^ is a classic coenzyme that exists in all species, and that plays a crucial role in oxidation–reduction reactions. It is also involved in the regulation of many cellular functions including inflammation, oxidative stress and differentiation. NAD^+^ declines with aging in various organs, and the reduction in NAD^+^ is possibly involved in the development of age-related cellular dysfunction in cardiorenal metabolic organs through the accumulation of inflammation and oxidative stress. Levels of NAD^+^ are regulated by the balance between its synthesis and degradation. CD38 is the main NAD^+^-degrading enzyme, and CD38 is activated in response to inflammation with aging, which is associated with the reduction in NAD^+^ levels. In this review, focusing on CD38, we discuss the role of CD38 in aging and the pathogenesis of age-related diseases, including cardiorenal metabolic disease.

## 1. Introduction

Aging is an important risk factor for the leading causes of mortality, and the incidence of age-related diseases, including cardiovascular disease, kidney disease and metabolic diseases such as metabolic syndrome, diabetes and non-alcoholic fatty liver, increases dramatically with age [1,2,3,4]. Aging and age-related diseases share some basic mechanisms that largely converge on inflammation and oxidative stress [5]. During the aging process, chronic low-grade inflammation and oxidative stress develop, which contributes to the age-related cellular dysfunction [5,6,7]. Inflammation is caused by mechanisms involving the overproduction of inflammatory cytokines and chemokines, and nuclear factor-kappa B (NF-κB) or inflammasome activation [8,9]. In addition, oxidative stress is caused by an imbalance between the increased production of reactive oxygen species (ROS) and decreased antioxidant systems [7]. Moreover, inflammation and oxidative stress are closely linked, leading to the further promotion of aging and age-related diseases [10].

NAD^+^ is a classic coenzyme that exists in all species and plays an important role in oxidation–reduction reactions. In addition, it is involved in the regulation of many cellular functions including inflammation, oxidative stress, circadian rhythms and differentiation [11]. NAD^+^ declines with aging in various organs of rodents and humans [12,13,14,15]. Therefore, the reduction in NAD^+^ is possibly involved in the development of age-related cellular dysfunction in cardiometabolic organs such as the heart, vascular tissue, liver, adipose tissue, kidney and skeletal muscle through the accumulation of inflammation and oxidative stress. Levels of NAD^+^ are regulated by the balance between its synthesis and degradation. Therefore, a reduction in NAD^+^-synthesizing enzymes and/or an enhancement of NAD^+^-degrading enzymes causes a reduction in NAD^+^. Nicotinamide phosphoribosyl transferase (NAMPT), the rate-limiting enzyme in the NAD^+^ biosynthetic salvage pathway, starts as nicotinamide (NAM), and decreases with aging in various organs such as the liver and skeletal muscle, and in white adipose tissue [12]. On the other hand, several enzymes such as NADases such as CD38 [16], poly (ADP-ribose) polymerases (PARPs) [17,18] and NAD^+^-dependent deacetylases (sirtuins) [19,20] can degrade or consume NAD^+^ during their catalytic processes. In particular, CD38 is recognized as the main NAD^+^-degrading enzyme [16], and an increase in CD38 with aging contributes to the reduction in NAD^+^ in mammalian tissues [21]. PARP1 activates for repair against oxidative DNA damage reactively, consuming NAD^+^ during this process [17,22]. In addition, sirtuins are activated in a NAD^+^-dependent manner in response to cellular stress including oxidative stress, inflammation and energy depletion, consequently leading to the maintenance of cellular homeostasis [19,20]. Although NAD^+^ is consumed upon sirtuin activation, sirtuins are not activated if the levels of NAD^+^ are not sufficient.

Among the enzymes involved in NAD^+^ synthesis and metabolism, numerous basic studies as described below have demonstrated that CD38 increases in response to inflammation with aging in the cardiovascular tissues, macrophages, liver, kidney and skeletal muscle, which is associated with the reduction in NAD^+^. Therefore, we focused on CD38 and discussed the role of CD38 in aging and the pathogenesis of age-related diseases, particularly cardiorenal metabolic diseases.

## 2. NAD^+^ Synthesis and Metabolic Cycle

Dietary nutrients or supplements are converted to NAD^+^ through endogenous biosynthetic pathways (Figure 1a).

### 2.1. De Novo Pathway and Preiss–Handler Pathway

The de novo pathway of NAD^+^ synthesis converts dietary tryptophan to quinolinic acid (QA) via enzymes including indoleamine-2,3-dioxygenase (IDO) in multiple steps (Figure 1a) [23]. Thereafter, QA converts to nicotinate mononucleotide (NAMN) via quinolinate phosphoribosyl transferase (QPRT) and induces NAD^+^ biosynthesis [23,24,25]. The Preiss–Handler pathway converts dietary nicotinic acid (NA) to nicotinate mononucleotide (NAMN) via nicotinate phosphoribosyltransferase (NAPRT) [26] (Figure 1a). NAMN derived from tryptophan or NA is catalyzed by nicotinamide mononucleotide adenylyl transferases (NMNATs) to convert nicotinic acid adenine dinucleotide (NAAD), which is amidated to NAD^+^ by NAD synthase (NADSYN) (Figure 1a) [27,28].

### 2.2. Salvage Pathway

NAD^+^ is recycled from nicotinamide (NAM), nicotinamide riboside (NR), NA and nicotinamide mononucleotide (NMN) in the salvage pathway in various cellular compartments such as the nucleus and mitochondria to maintain cellular NAD^+^ levels (Figure 1a) [29,30]. NAM is also recycled into NMN by NAMPT, which catalyzes the rate-limiting reaction in the salvage pathway (Figure 1a) [31]. In addition, NR is imported to the cells thorough equilibrative nucleoside transporters (ENTs) and transformed to NMN by nicotinamide riboside kinases 1 and 2 (NRK1/2) (Figure 1a) [32]. Thereafter, NMN is adenylylated by nicotinamide mononucleotide adenylyltransferases (NMNATs), which include isoforms 1-3, to generate NAD^+^ in the nucleus, Golgi, mitochondria, endolysosome and cytosol (Figure 1a) [33,34,35,36]. Then, NAD^+^ is converted to NAD^+^ phosphate (NADP^+^) by NAD^+^ kinases (NADKs) (Figure 1a). Moreover, NAD^+^ and NADP^+^ turn into their reduced forms NADH and NADPH, respectively, in several metabolic reactions (Figure 1a).

Extracellular NAD^+^ is converted by CD38, its homologue CD157/bone marrow stromal cell antigen (BST)-1 and CD73 (Figure 1b) [37,38]. CD38 can convert NAD^+^ or NMN to NAM, and CD157/BST-1 can convert NAD^+^ or NR to NAM (Figure 1b). In addition, CD73 can convert NAD^+^ to NMN, and CD73 can re-convert from NMN to generate NR [39,40]. NAM is imported into cells; however, its transporter is still unknown [41,42]. NR can enter the cell through equilibrative nucleoside transporters [43,44]. NMN converts to NR, which allows it to be transported into the cells. NMN itself can be taken up via Slc12a8 [45]. Thus, the metabolites, NAM, NR and NMN, released by these ectoenzymes, including CD38, CD157 and CD73, can enter the cell and be used to produce intracellular NAD^+^.

On the other hand, intracellular NAD^+^ is metabolized by NAD^+^-consuming enzymes including CD38 [16], sterile alpha and TIR motif containing 1 (SARM1) [46,47,48], PARP1 [17,18] and sirtuins such as Sirt1 and Sirt3 [19,20]. The enzymatic reaction by CD38 produces NAM as a byproduct that can be recycled back into the NAD^+^ salvage pathway, and this generates cyclic ADP-ribose (cADPR) or ADP-ribose (ADPR) (Figure 1b). In addition, SARM1 is one of the NAD^+^ consumers in neurons, and the dimerization of the TIR domain cleaves NAD^+^ into cADPR, ADPR and NAM (Figure 1b) [46,47,48]. Sirtuins consume NAD^+^ during the reaction and the acetyl group is removed from the acetylated protein. DNA damage activates PARP1-mediated NAD^+^ consumption during PARylation, in which the ADPR chain is added onto the target proteins. In these reactions by sirtuins and PARP1, NAM is produced as a byproduct of NAD^+^ consumption.

## 3. Biology of CD38

CD38 is a multifunctional enzyme that is ubiquitously distributes in mammalian tissues, which uses NAD^+^ as a substrate to produce cADPR and ADPR [49]. CD38 has NAD^+^ glycohydrolase [50], ADP-ribosyl cyclase and cADPR hydrolase enzymatic activities [51,52]. Function of CD38 as NAD^+^ glycohydrolase, a NADase, catalyzes the reaction with NAD^+^, producing ADPR and NAM (Figure 2a) [49,50]. In addition, CD38 acts as ADP-ribosyl cyclase for cADPR synthesis from NAD^+^, and it generates NAM (Figure 2a) [51]. Furthermore, CD38 functions as cADPR hydrolase for the hydrolysis of cADPR to form ADPR (Figure 2a) [51,52,53]. Moreover, cADPR plays as role as an important second messenger, regulating intracellular Ca^2+^ homeostasis in several cells [54,55]; cADPR functions as the trigger for Ca^2+^ release via ryanodine receptor (RyR) activation, located in the endoplasmic reticulum (ER) (Figure 2b). In addition to NADase, at acidic conditions and in the presence of suitable amounts of NA, CD38 can catalyze a nucleobase exchange reaction between NA and the NAM moiety of NADP, producing nicotinic acid adenine dinucleotide phosphate (NAADP) [54,55,56,57]. NAADP elicits Ca^2+^ release from the two-pore channel (TPC) receptor located in acidic lysosomes or endolysosomes, or from the RyR on ER, having implications for intra-cellular Ca^2+^ homeostasis [58,59,60,61,62]. (Figure 2b). In addition to CD38, NAADP is generated by NADPH oxidase or dual NADPH oxidase (DUOX) from NAADPH. (Figure 2b) [63,64].

Basic research using CD38 (−/−) mice has clearly shown that CD38 is the main NADase in mammalian tissues and is involved in the degradation of NAD^+^ [16]. In fact, the levels of tissue NAD^+^ in CD38 (−/−) mice were around 10- to 20-fold higher than in wild-type mice [16]. Thus, CD38 is closely involved in regulating cellular and tissue NAD^+^ homeostasis. CD38 situates in two opposite membrane orientations, with extracellular and cytosolic catalytic sites, as type II and III, respectively [65,66]. The great majority of CD38 is located in the plasma membrane, having a type II membrane orientation [67], degrading not only NAD^+^, but also circulating the NAD^+^ precursor NMN, prior to transportation to the inside of the cells for NAD^+^ biosynthesis [21]. Thus, the role of type II CD38 is maintain NAD^+^ homeostasis in the extracellular environment by regulating both NAD^+^ and the precursors of NAD^+^ synthesis. In addition, type III CD38 exists in the membranes of intracellular organelles including the nucleus, mitochondria, endoplasmic reticulum, endolysosomes and lysosomes and also on cell membranes facing the cytosol, and plays an important role in regulating the level of intra-organelle NAD^+^, cADPR and Ca^2+^ [53,65,66,68,69]. Further, type II CD38 paracrinally supplies a neighboring concentrative nucleoside transporter (CNT)-and RyR-expressing cells with cADPR, regulating intracellular Ca^2+^ mobilization and its related cellular functions, in several cells such as smooth muscle cells, 3T3 murine fibroblasts, hippocampal neurons and human hemopoietic stem cells [70,71].

The promoter region of the human CD38 gene, situated at chromosome 4, is regulated by several transcriptional factors including NF-κB, activator protein-1 (AP-1), retinoid X receptor (R×R), liver X receptor (L × R) and signal transducer and activator of transcription (STAT) [72,73,74,75,76]. Tumor necrosis factor-α (TNF-α) activates the mitogen-activated protein kinases (MAPKs) such as the extracellular signal-regulated kinase (ERK), p38 and c-Jun N-terminal kinase (JNK). Although p38 and JNK upregulates CD38 expression through the activation of NF-κB and AP-1, ERK and p38 may be involved in stabilizing the CD38 transcripts [77,78]. In addition to TNF-α, CD38 expression and its activation is induced by inflammatory cytokines such as interferon-γ (IFN-γ) and lipopolysaccharide (LPS) and [72,73,74,76,79,80], resulting in a subsequent cellular NAD^+^ decline (Figure 3). The pro-inflammatory state observed during the aging process may be linked to cytokine-induced CD38 expression and NAD^+^ decline in age-related organs/tissues, and is referred to as “inflammaging” [81] (Figure 3). In addition, ROS are also involved in CD38 activation through several mechanisms. In coronary endothelial cells, ROS derived from NADPH oxidase activation that is activated by endothelin-1 (ET-1) promotes CD38 internalization and exposes the active enzymatic site to its substrate NAD^+^ or NADP^+^ through membrane raft clustering [82] (Figure 3). CD38 has 12 highly conserved cysteine (Cys) residues in the extracellular domain. Oxidation of Cys molecules such as Cys 119 and Cys 201 leads to the formation of several disulfide bonds, possibly changing the CD38 protein conformation to trigger CD38 internalization [83,84]. Park et al. also reported that ROS derived from NADPH oxidase activates type III CD38 by forming a disulfide bond between Cys164 and Cys177 in the early endosomes of lymphokine-activated killer (LAK) cells treated with interleukin-8 (IL-8) [85] (Figure 3). On the other hand, other report has shown that high expression of CD38-induced NAD^+^ decline increases oxidative stress through downregulation of antioxidation-related genes [86]. Ferroptosis is recognized as a form of cell death driven by iron-dependent lipid peroxidation, and it may be involved in cellular aging [87,88]. Ma et al. demonstrated that in bone marrow-derived macrophages from aged mice, increased expression of CD38 may trigger oxidative degradation of dihydrofolate reductase (DHFR) through sulfonation of the Cys 7 residue, increasing the susceptibility of ferroptosis [89]. Thus, CD38 and oxidative stress interact and may be involved in cellular aging.

CD38 may be involved in the regulation of the NAD^+^ pools. Enhanced CD38 activity may lead to a severe NAD^+^ decline. CD38-induced NAD^+^ decline decreases the activity of the NAD^+^-dependent enzymes, sirtuins such as Sirt1 and Sirt3, which are closely involved in the regulation of cellular functions including energy homeostasis, inflammation, oxidative stress, autophagy, mitochondrial function and aging, consequently leading to metabolic dysfunction, promoting the aging process and the progression of age-related diseases, as described below.

## 4. Role of CD38 on Aging, Metabolic Dysfunction, High-Fat-Diet-Induced Obesity and Insulin Secretion

The impact of CD38 on the pathogenesis of aging and age-related disease has been shown by CD38 (−/−) mice and pharmacological CD38 inhibition. Camacho-Pereira et al. clearly showed that the expression and activity of CD38 in the liver, adipose tissue and skeletal muscle of mice, increased with aging and was required for the age-related NAD^+^ decline [21] (Figure 4a). The study using CD38 (−/−) mice showed that CD38 plays a crucial role in the pathogenesis of mitochondrial dysfunction and its related metabolic dysfunction through NAD^+^ decline and decreasing Sirt3 activity (Figure 4a). A specific thiazoloquin(az)olin(on)e CD38 inhibitor, 78c, prevents age-related NAD^+^ decline and improves physiological and metabolic dysfunction such as glucose intolerance, muscle dysfunction, reduction in exercise capacity and cardiac dysfunction in the aging mouse [90] (Figure 4a). In addition, 78c increases the levels of NAD+, leading to the activation of pro-longevity factors including sirtuins and AMP-activated kinase (AMPK), and suppression of the pathway that negatively affect health span, such as the mechanistic target of rapamycin-S6 kinase (mTOR-S6K) [90] (Figure 4a). Furthermore, the CD38 inhibitor 78c increases the lifespan and health span of naturally aged mice with a 10% increase in median survival, accompanied by improvements in exercise performance, endurance and metabolic function [91] (Figure 4a). In addition, in humans, CD38 mRNA expression in white adipose tissue derived from elderly subjects has been shown to be significantly elevated, compared with that of younger subjects [92]. Furthermore, CD38 activity in the blood has been shown to be significantly positively correlated with age [92].

CD38 (−/−) mice are protected against high-fat-diet-induced obesity, increased lipid droplets in the hepatocytes and skeletal muscle, and glucose intolerance by enhancement of energy expenditure via increased mitochondrial biogenesis, which is related to the activation of Sirt1 and peroxisome proliferator-activated receptor γ coactivator 1α (PGC-1α) [93] (Figure 4a). A high-fat/high-sucrose diet in wild-type mice has been shown to result in exercise intolerance, a decrease in metabolic flexibility and glucose intolerance [94]. Loss of CD38 by genetic mutation protects mice against diet-induced metabolic derangement [94] (Figure 4a). In addition, CD38 deficiency protect against high-fat-diet-induced obesity through the suppression of adipogenesis- and lipogenesis-related transcriptional factors and target genes such as sterol regulatory element binding protein-1c (SREBP1c), CCAAT/enhancer binding protein (C/EBP), peroxisome proliferator-activated receptor γ (PPARγ) and fatty acid synthase, by Sirt1 activation [95] (Figure 4a). CD38 inhibition by flavonoids including apigenin and quercetin also improves glucose homeostasis and promotes fatty acid oxidation in the liver of mice with high-fat-diet-induced obesity, possibly through the activation of sirtuins such as Sirt1 and Sirt3 [96] (Figure 4a). In addition, CD38 plays a crucial role in the pathogenesis of high-fat-diet-induced non-alcoholic fatty liver disease (NAFLD). CD38 deficiency suppresses the initiation and progression of NAFLD by reducing lipid accumulation and suppressing oxidative stress via the activation of sirtuins such as Sirt1 and Sirt3 [97] (Figure 4a). Thus, suppression of CD38 may be beneficial for protection against aging, a shortened health span, age-related diseases, high-fat-diet-induced obesity and metabolic dysfunction (Figure 4a).

Accumulation of senescent cells (the senescence-associated secretory phenotype; SASP) with a secretory phenotype promotes chronic low-grade inflammation, leading to age-related dysfunctions [98] (Figure 4b). Although senescent cells show no high expression of CD38, the SASP factors including inflammatory cytokines secreted by senescent cells induced CD38 expression and its activation in non-senescent cells, such as proinflammatory M1-like macrophages, not native or M2 macrophages [99] (Figure 4b). Accumulating higher CD38 expressed M1-like macrophages in tissues such as the white adipose tissue and liver during chronological aging causes to NAD^+^ decline in the tissues, further promoting the aging process and metabolic dysfunction [100] (Figure 4b). In addition, increased expression of CD38 has been observed in endothelial cells and macrophages in response to the SASP factors, possibly leading to the initiation and progression of cardiovascular disease, which is related to aging [101] (Figure 4b).

The role of CD38 in insulin production or secretion is controversial. Okamoto and colleagues reported that the genetic disruption of CD38 in mice of a mixed genetic background produced severe impairment of glucose tolerance [102]. It was shown that impaired β-cell insulin secretion was related to the absence of cADPR, a by-product of the CD38 catalytic reaction, to mobilize Ca^2+^ from the endoplasmic reticulum in response to glucose [102]. However, other reports have shown that ablation of CD38 followed by 12 backcrosses to the defined C57BL/6J background did not demonstrate impairment of glucose tolerance [103]. Thus, it is possible that the influence of genetic background rather than CD38 deficiency alone is involved in the impairment of glucose tolerance and β-cell function. In addition, in insulitis-free NOD-SCID mice with a disrupted CD38 gene, CD38 was not essential for normal glucose clearance of intraperitoneal-injected glucose. Even though CD38 deficiency showed no impaired response to acute glucose challenge, CD38-deficient β-cells might be more sensitive to the induction of apoptosis [104]. Further studies are necessary to clarify the role of CD38 in insulin secretion from pancreatic β-cells.

## 5. Role of CD38 on Cardiovascular Disease and Kidney Disease

Cardiovascular aging leads to changes in the heart, and vascular structure and function including progression of myocardial remodeling, left ventricular hypertrophy, vulnerability to ischemia, reduced systolic and diastolic function, atherosclerosis and hypertension, resulting in high mortality related to heart failure and ischemic heart disease.

### 5.1. CD38 Gene Ablation or Inhibition Protects the Heart

Using heart disease animal models, CD38 deficiency illustrates the beneficial effect in terms of the protection against heart injuries via NAD^+^-dependent enzymes and the activation of sirtuins. Oxidative stress is closely linked to the pathogenesis of cardiovascular disease [105]. Myocardial ischemia/reperfusion (I/R) injury leads to oxidative damage in the cardiac muscle via the obstruction of blood flow to the myocardium and the reperfusion of ischemic areas of the heart thereafter [106]. Previous basic in vitro and in vivo research has demonstrated that CD38 deficiency plays a crucial role in the protection against oxidative stress. An in vitro study of mouse embryonic fibroblasts (MEFs) collected from CD38 (−/−) mice showed that they were significantly resistant to oxidative stress and hypoxia/reoxygenation-induced cell injury, compared with wild-type MEFs [107]. In addition, CD38 deficiency protects the heart against I/R injury in mice by suppressing the generation of ROS in myocardial cells, via the enhancement of antioxidative enzymes including catalase and SOD2 induced via the Sirt1/FoxO1 and three-pathway activation [108]. The CD38-mediated suppression of Ca^2+^ overload in myocardial cells also contributes to cardiac protection [108].

Furthermore, in a study involving the collection of myocardial biopsy specimen from Tetralogy of Fallot patients and hypoxia/ischemia-related myocardial injury models including myocardial infarction and a thermal burn experimental mice model, CD38 was upregulated in injured cardiac tissue [109]. Rab7 and its adaptor protein pleckstrin homology and run domain-containing M1 (PLEKHM1) are important molecules for the formation of autolysosomes via the fusion of autophagosomes and lysosomes. Overexpression of CD38 downregulates the expression of Rab7 and PLEKHM1 in the cardiac tissue of burn model mice [109]. As the mechanism, Sirt1, activated by an increase in NAD^+^ resulting from CD38 deficiency, induces Rab7 transcription through the activation of forkhead box protein O1 (FoxO1) via deacetylation [109]. On the other hand, CD38 inhibits PLEKHM1 expression; however, this is not dependent on the NAD^+^-Sirt1 pathway, and the mechanism remains unclear. The loss of Rab7/PLEKHM1 suppresses autophagic flux through the impairment of the fusion of autophagosomes and lysosomes, leading to the accumulation of abnormal mitochondria and increased apoptosis, and consequent cardiac dysfunction under hypoxic/ischemic conditions.

Over-nutrition such as a high-fat diet may result in cardiac dysfunction through metabolic-related changes including lipid accumulation, mitochondrial dysfunction and increased oxidative stress. A previous report has shown that CD38 deficiency may protect the heart against high-fat-diet-induced oxidative stress through the enhancement of antioxidative stress via the Sirt3-mediated activation of the forkhead box protein O3 (FoxO3)/superoxide dismutase2 (SOD2) signaling pathway in mice [110].

Left ventricular hypertrophy is an important predictor of adverse cardiovascular outcomes and a risk factor for heart failure [111], and it is regulated by various factors, such as angiotensin-II (Ang-II), ET-1, catecholamines and TNF-α [112]. Among them, Ang-II may induce pressure overload, resulting in wall thickening and left ventricular concentric hypertrophy as a compensatory mechanism to maintain the ventricular ejection fraction under situation of increased peripheral resistance [112]. Ang-II promotes cardiac hypertrophy by activating calcineurin-nuclear factor of the activated T cells 4 (NFATc4) pathway and enhancing oxidative stress. CD38 deficiency protects the heart against hypertrophy via the activation of Sirt3 [113]. The activation of Sirt3, by increased levels of NAD^+^ induced by CD38 deficiency, activates AMPK, which suppresses Ang-II-induced cardiac hypertrophy by inhibiting the AKT-glycogen synthase kinase3β (GSK3β)-NFATc pathway [113]. In addition, the increased Sirt3 activities promote the expression of antioxidative enzymes such as SOD2 and catalase via the FoxO3 pathway activation, leading to the attenuation of Ang-II-induced cardiac hypertrophy via a reduction in oxidative stress [113].

d-galactose promotes cardiomyocytes senescence and ROS production, and the expression of CD38 is increased in the senescent cardiomyocytes of old mice, while the expression of NAMPT and Sirt1 is decreased [114]. CD38 knockdown can attenuate myocardial cell senescence and oxidative stress by d-galactose, while the Sirt1-specific inhibitor reverses the effects of CD38 deficiency on cellular senescence and oxidative stress. Therefore, the CD38/Sirt1/NAD^+^ axis may play crucial roles in d-galactose-induced myocardial cell aging [114].

In addition to sirtuin activation, other cardioprotective effects of CD38 deficiency have been proposed. The elevation of serum testosterone levels in male CD38 (−/−) mice enhances cardiac function through upregulating major genes related to myocardial contraction, including RyR2, sarcoplasmic reticular Ca^2+^ ATPase (SERCA2) and Na^+^/Ca^2+^-exchanger protein 1 (NCX1), and α myosin heavy chain (α-MHC) [115]. Another report demonstrated that CD38 inhibition improved exercise performance, possibly through increasing NAD^+^ [116]. In addition, CD38 knockout, catalytically inactive CD38 and treatment with an antibody for CD38 (Ab68) in wild-type mice contributed to a decrease in basal heart rate, an increase in heart rate variability and altered calcium handling, protecting mice against developing catecholamine-induced ventricular arrhythmias [116]. Moreover, the chronic stimulation of β-adrenergic signaling by isoproterenol activates an NAADP synthesizing enzyme and CD38 to increase intracellular Ca^2+^ by elevating NAADP and cADPR levels, respectively [55]. The sustained increase in intracellular Ca^2+^ induced by isoproterenol led to cardiac dysfunction, fibrosis and hypertrophy in wild-type mice, whereas CD38 (−/−) mice were resistant to isoproterenol-induced cardiac injury [55]. Thus, cADPR-mediated Ca^2+^ signaling induced by CD38 activation plays a critical role in isoproterenol-induced cardiac hypertrophy and fibrosis.

Interestingly, Duchenne muscular dystrophy (DMD), a progressive muscle disease, has two important deleterious features of one of the pathophysiological mechanisms [117]. One is Ca^2+^ dysregulation linked to Ca^2+^ influxes associated with ryanodine receptor hyperactivation; the other is a muscular NAD^+^ reduction. Deletion of CD38 in mdx mice, a model of DMD, led to an improvement in heart function/structure and skeletal muscle performance, related to a reduction in cellular inflammation and senescence, through restored NAD^+^ levels and reduced pathological spontaneous Ca^2+^ activity [117]. A summary of this section is shown in Table 1.

### 5.2. CD38 Gene Ablation or Inhibition of Its Activation Protects Vascular Tissue and Cells

Ang-II induces intracellular ROS production and mitochondrial dysfunction, and increases the generation of small extracellular vesicles (sEVs), which are critical for vascular smooth muscle cell (VSMC) senescence in hypertension. The sEVs, including exosomes, are released from many cell types and play crucial roles in cell-to-cell and interorgan communication. Cellular senescence contributes to promoting the aging process through the release of soluble factors such as the SASP, which includes particularly senescence-associated sEVs (SAsEVs). CD38 deficiency attenuated Ang-II infusion-induced hypertension and vascular remodeling [118]. As the mechanism, CD38 deficiency alleviated VSMC senescence by inhibiting the biogenesis, secretion and internalization of SAsEVs in Ang-II-stimulated VSMCs, in relation to maintaining mitochondrial homeostasis and mitophagy activation through the activation of Sirt1- and Sirt3-mediated restoration of lysosome function [118].

In addition, the endothelial nitric oxide synthetase (eNOS) substrate NADPH was depleted in the endothelial cells of the postischemic heart of an I/R animal model, which occurred via the high expression and activation of the NAD(P)ase function of CD38, triggering eNOS dysfunction and overproduction of ROS due to uncoupling [119,120,121,122]. CD38 inhibition by α-NAD supplementation, luteolinidin administration and genetic deletion restored normal endothelial vasodilation. This restoration of endothelial function including nitric oxide (NO) production by CD38 inhibition is important, and could possibly lead to the enhanced recovery of cardiac contractile function and decreased infarction [119,120,121,122]. Thus, a decrease in NAD^+^ by CD38 activation affects the suppression of sirtuins and dysfunction of eNOS, and consequently, vascular injury including SMC senescence or postischemic endothelial dysfunction.

In addition to NAD^+^ depletion, the increased expression of CD38 in diabetes involves the release of intracellular cADPR-mediated Ca^2+^ and mitochondrial damage, resulting in Nlr family pyrin domain-containing 3 (Nlrp3) inflammasome activation, VSMC proliferation and collagen I synthesis [123]. A summary of this section is shown in Table 2.

### 5.3. CD38 Gene Ablation or Inhibition of Its Activation Promotes Coronary Atherosclerosis

Both cADPR and NAADP generated through CD38 activation are recognized as important potent intracellular Ca^2+^-mobilizers, as described above. In addition, the intracellular transport and fusion of lysosomes with autophagosomes requires an increased cytosolic Ca^2+^ concentration. Therefore, a decrease in cADPR or NAADP may impair autophagic flux via a reduction in the intracellular calcium content. Several studies by Li PJ and colleagues reported that CD38 gene ablation or inhibition of its activation promotes coronary atherosclerosis.

CD38 deficiency causes dysregulation of autophagic flux, reducing collagen I degradation in coronary artery myocytes and leading to its deposition around the coronary artery and the progression of atherosclerosis [124]. In addition, dysfunction of autophagy flux including lysosomal V-ATPase inhibition, by CD38 gene deletion, resulted in Nlrp3 inflammasome formation and activation, and consequent coronary artery inflammation and remodeling [125]. Furthermore, endocytosed oxidized low-density lipoprotein (LDL) is transported into lysosomes, in which the cholesterol ester is hydrolyzed to free cholesterol. The CD38 enzymatic product, NAADP, functions as a Ca^2+^ messenger to release Ca^2+^ from lysosomes. The local Ca^2+^ increase activates free cholesterol transporters (such as Niemann–Pick type C1) and promotes the discharge of cholesterol from lysosomes [126]. A deficit in NAADP-mediated Ca^2+^ release from lysosomes caused by CD38 deficiency leads to free cholesterol accumulation in lysosomes, and facilitates atherosclerosis [126].

The CD38 enzymatic products cADPR and NAADP and associated Ca^2+^ signaling are closely involved in the control of nuclear factor erythroid-2-related factor 2 (Nrf2) activation through Nox4-dependent ROS production. Conversely, CD38 deficiency suppresses Nrf2 activation [127]. Since Nrf2 plays a critical role in protecting cells against oxidative stress, Nrf2 may be a novel therapeutic target for preventing atherosclerosis [128]. CD38 deficiency suppresses Nrf2 activation by reducing Nox4-induced ROS production, and decreasing the expression of contractile marker calponin, smooth muscle protein 22-α (SM22α) and α-smooth muscle actin (α-SMA), but increasing the expression of the smooth muscle cell dedifferentiation marker, vimentin, enhancing cell proliferation and promoting atherosclerosis [127].

Thus, from the results of basic research, it appears that CD38 may play a dual role, relating to either anti-atherosclerosis or promoting atherogenesis, depending on the change in sirtuin activity or cADPR- and NAADP-mediated Ca^2+^ mobilization. A summary of this section is shown in Table 2.

### 5.4. CD38 Plays a Crucial Role in Vasoconstriction

CD38 contributes to the regulation of Ang-II, ET-1 and norepinephrine-induced vasomotor tone in the systemic and renal vasculature through the generation of cADPR, a second messenger that activates ryanodine receptors to release Ca^2+^ from the sarcoplasmic reticulum in VSMCs [129,130].

In addition to Ca^2+^ release via CD38-ADP ribosyl cyclase, ROS production via the interplay between CD38 and NADPH oxidase also plays a crucial role in the regulation of vascular tone [131]. The NADP^+^ is consumed by CD38, consequently contributing to depletion of NADP(H) pool, which is the substrate of NADPH oxidase. ROS produced by NADPH oxidase leads to vasoconstriction. In muscarinic type 1 receptor activation by oxotremorine, CD38 is essential for intracellular ROS production by the cADPR/Ca^2+^-mediated activation of the NADPH oxidase subunit, Nox4 [131]. However, extracellular ROS produced in an independent manner via the CD38/cADPR/Ca^2+^ pathway is derived from another NADPH oxidase subunit, Nox1. Extracellular ROS further augments CD38-mediated intracellular ROS production in coronary artery myocytes (CAMs), resulting in vasoconstriction in the coronary arteries [131]. In addition, Moss NG et al. showed that the thromboxane mimetic U-46619 causes renal vasoconstriction via the stimulation of the CD38/cADPR/Ca^2+^ pathway, and renal vasoconstriction is attenuated by the antioxidant SOD mimetic [132]. Thromboxane prostanoid receptors stimulate CD38 and ROS production, each of which contributes to vasoconstriction. In addition to the independent effects on renal vasoconstriction, an interaction exists in which ROS activates CD38 to cause part of its action.

### 5.5. Role of CD38 on Kidney Disease

Chronic kidney disease, such as diabetic kidney disease and hypertensive glomerulosclerosis, is recognized as an important cause of end-stage kidney disease. Aging is one of the risk factors for the progression of renal function decline in both diabetic kidney disease (DKD) and hypertensive glomerulosclerosis.

Previously we reported that high-glucose-induced CD38 is responsible for the decreased NAD^+^/NADH ratio, and the suppression of Sirt3 activity further results in mitochondrial oxidative stress characterized by elevated mitochondrial antioxidative enzymes, acetylated SOD2 and isocitrate dehydrogenase 2 (IDH2) in renal tubular cells [133]. In addition, treatment with apigenin, a CD38 inhibitor, clearly improved diabetic-induced kidney injury through the restoration of Sirt3 activation via an increased NAD^+^/NADH ratio and improvement of mitochondrial oxidative stress, particularly in tubular cells [133]. On the other hand, several studies by Li PJ and colleagues have demonstrated that the normal expression of CD38 plays a crucial role in the regulation of the integrity of podocyte structure and function, and that ablation or deletion of CD38 gene expression leads to epithelial mesenchymal transition (EMT) [134] and impairment of autophagic flux by inhibition of autolysosome formation [135]. In addition, in podocytes of CD38-deficient mice, podocin expression was markedly decreased, and deoxycorticosterone acetate (DOCA)/high salt treatment further decreased podocin expression [134]. The desmin, an intermediate filament protein and a specific podocyte injury marker, was increased in CD38-deficient mice, and DOCA/high salt treatment further enhanced desmin expression [134]. Thus, CD38 is required for normal podocyte structure and function, and CD38 deficiency induces glomerular damage and sclerosis through podocyte injury. Thus, regarding the role of CD38 in renal structure and function, it is important that its expression level and function is in an appropriate balance, that is neither excessive nor insufficient, in order to maintain homeostasis.

Elderly people are at high risk of acute kidney injury (AKI), and they show poor outcomes in terms of complications, including progression to chronic kidney disease and end-stage renal disease or mortality. AKI may be caused by a variety of pathogenic factors, such as sepsis [136,137]. In the pathogenesis of sepsis-induced AKI, LPS has been identified as the major component [137]. A previous report demonstrated that LPS upregulates CD38 expression, and the knockdown or blockade of CD38 may inhibit macrophage M1 polarization accompanied by a reduction in NF-κB signaling activation in a cultured macrophage and in an LPS-induced AKI mouse model [74]. On the other hand, Li et al. showed that the increase in the toll-like receptor 4 (TLR4) expression in CD38 (−/−) mice may promote the production of IFN-γ, contributing to more severe AKI in LPS-induced sepsis [138]. Thus, the effects of CD38 blockade in bacterial-sepsis-induced AKI in experimental models is contradictory. Further studies are necessary to elucidate the role of CD38 on the initiation and progression of AKI. A summary of this section is shown in Table 3.

## 6. Conclusions and Prospectives

In this review, we summarized the role of CD38 in the pathogenesis of aging and cardio–renal–metabolic diseases, primarily from reports of basic research using animal models. NAD^+^ plays an important role in an array of cellular processes such as mitochondrial function and metabolism, oxidative stress, inflammation, cell signaling, cell division and DNA repair. Therefore, dysregulation of NAD^+^ metabolism is related to aging and age-related metabolic derangement and multiple diseases. CD38 is the main NADase in mammalian tissues, and the role of CD38-activation-related NAD^+^ decline has been investigated in relation to the pathogenesis of the aging process, metabolic dysfunction and age-related diseases. Numerous studies in basic research have shown that the inhibition of CD38 may be a potential therapeutic target for aging and cardiorenal metabolic disease.

Conversely, several studies have reported that inhibition of CD38 may lead to the exacerbation of the pathological conditions such as insulin secretion, and cardiovascular and kidney disease. This discrepancy may be based on the feature of CD38, functioning not only as the regulator of NAD^+^, but also the regulator of intracellular Ca^2+^ homeostasis via cADPR and NAADP. Therefore, the role of CD38 in the pathogenesis of several diseases requires further evaluation, both in terms of NAD^+^ metabolism and intracellular Ca^2+^ homeostasis.

Could CD38 be a potential therapeutic target for human aging and cardiorenal metabolic disease? In clinical practice, CD38-targeted therapy using CD38 monoclonal antibodies has been performed in patients with hematologic malignancies such as multiple myeloma (MM), which highly expresses CD38 on their malignant cells [139,140,141,142,143]. CD38 is expressed not only on hematological malignant cells in MM, but also on some solid tumor cells. Interestingly, the role of CD38 in solid tumor cells has shown conflicting data, indicating that CD38 may be involved in both tumor progression and suppression depending on tumor type [144,145,146,147]. In addition, CD38 is expressed on tumor-related immunosuppressive cells, including regulatory T and regulatory B cells and myeloid-derived suppressor cells [139,148]. Therefore, treatment with CD38 antibodies may exhibit anti-tumor activity by releasing T-cell suppression via inhibition of such immunosuppressive cells, and may have therapeutic potential for solid tumors beyond hematologic malignancies such as MM. Follow-up data regarding the cardiorenal outcomes or metabolic changes including diabetes on the patients treated with CD38 antibodies may be useful to clarify the role of CD38 on cardiorenal metabolic disease in humans. However, since there are no previous reports showing such clinical data, the role of CD38 in human aging and the pathophysiology of cardiorenal metabolic disease remains unclear.

Thus, although structural variations of the molecule or specific association with different molecules according to the organ, tissue or cell may produce different functional effects, respectively; CD38 has both physiological and pathological roles, and great potential as one of the candidate molecules for anti-aging. However, further studies are necessary to clarify whether the CD38 may be a therapeutic target for human aging and age-related diseases, including cardiorenal metabolic disease.

## Figures and Tables

**Figure 1 cells-12-00595-f001:**
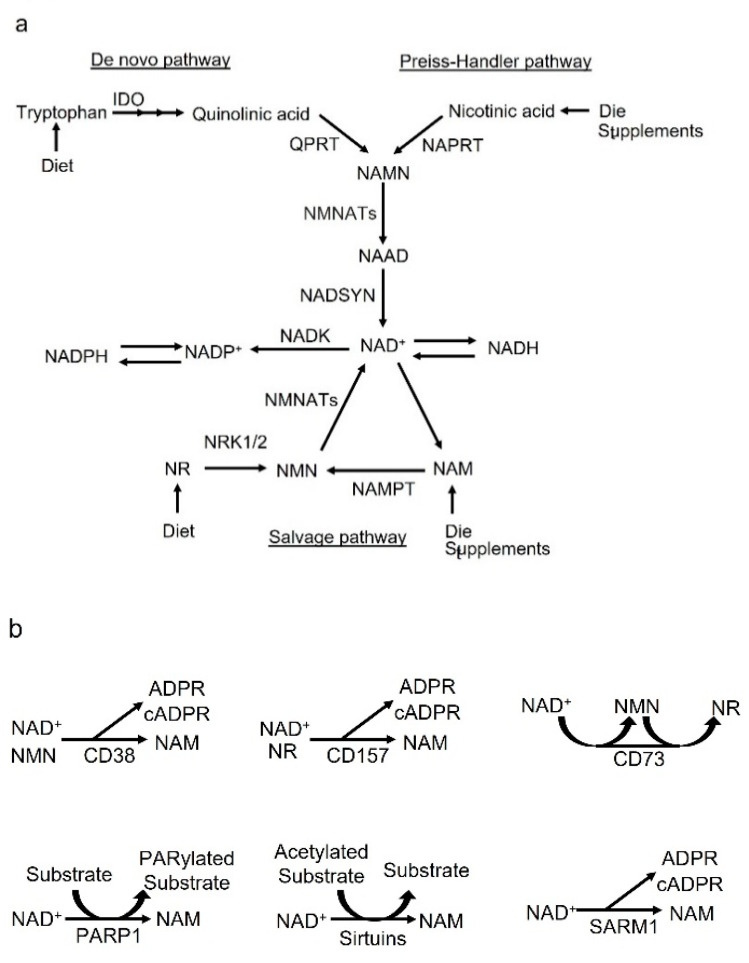
(**a**). NAD^+^ biosynthesis pathway. The de novo pathway: The conversion of tryptophan to quinolinic acid occurs through a series of enzymatic steps including IDO, a rate-limiting enzyme that catalyzes the first step in this pathway. QPRT catalyzes the conversion of quinolinic acid to NAMN. The Preiss–Handler pathway: NA converts to NAMN, catalyzed by NAPRT. NAMN is derived from tryptophan or NA is catalyzed by NMNATs to convert NAAD, which is amidated to NAD^+^ by NADSYN. The salvage pathway: NAM is converted to NMN by the rate-limiting enzyme NAMPT. NR is transformed to NMN by NRK1/2. CD73 can re-convert from NMN to generate NR. NMN is adenylylated by NMNATs to generate NAD^+^. NAD^+^ is converted to NADP^+^ by NADKs. Dietary nutrients or supplements are converted to NAD^+^ through endogenous biosynthetic pathways. (**b**). NAD^+^-consuming enzymes. CD38 can convert NAD^+^ or NMN to NAM, and CD157/BST-1 can convert NAD^+^ or NR to NAM. NAD^+^ is consumed by NAD^+^-consuming enzymes including CD38 and PARP1, and sirtuins such as Sirt1 and Sirt3. These enzymatic reactions produce NAM as a byproduct, which can be recycled back into the NAD^+^ salvage pathway, and produce cADPR or ADPR. In addition, SARM1 is an important NAD^+^ consumer in neurons. The dimerization of the TIR domain cleaves NAD^+^ into cADPR, ADPR and NAM. Sirtuins mediate NAD^+^ consumption during deacetylation, in which the acetyl group is removed from the acetylated protein. DNA damage activates PARP1-mediated NAD^+^ consumption during PARylation, in which the ADP-ribose chain is added onto the target proteins. In these reactions by the sirtuins and PARP1, NAM is generated as a byproduct from NAD^+^ consumption.

**Figure 2 cells-12-00595-f002:**
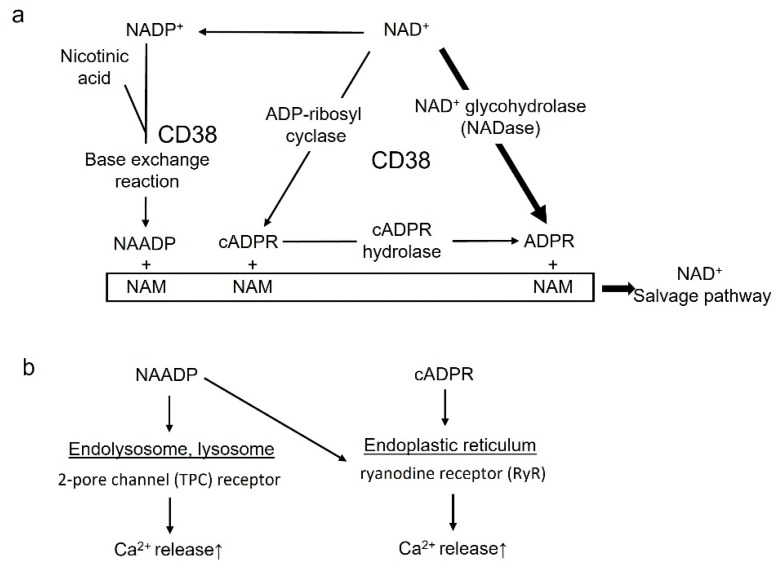
(**a**). Enzymatic activity of CD38. CD38 metabolizes NAD^+^ to ADPR through its glycohydrolase (NADase) activity or cADPR through its cyclase activity. The main activity of CD38 is recognized as a NADase. In addition, cADPR is converted to ADPR via cADPR hydrolase. In the presence of NA, CD38 can mediate a base-exchange reaction, in which NADP^+^ is converted to NAADP. NAM is generated as a byproduct from NAD^+^ consumption, recycling NAM back into the NAD^+^ salvage pathway. (**b**). NAADP and cADPR as the second messengers of Ca^2+^ release NAADP elicits Ca^2+^ release from the two-pore channel (TPC) receptor situated in acidic lysosomes or endolysosomes, or via the activation of the ryanodine receptor (RyR), situated in the endoplasmic reticulum. cADPR serves as the trigger for Ca^2+^ release via the activation of the RyR, situated in the endoplasmic reticulum.

**Figure 3 cells-12-00595-f003:**
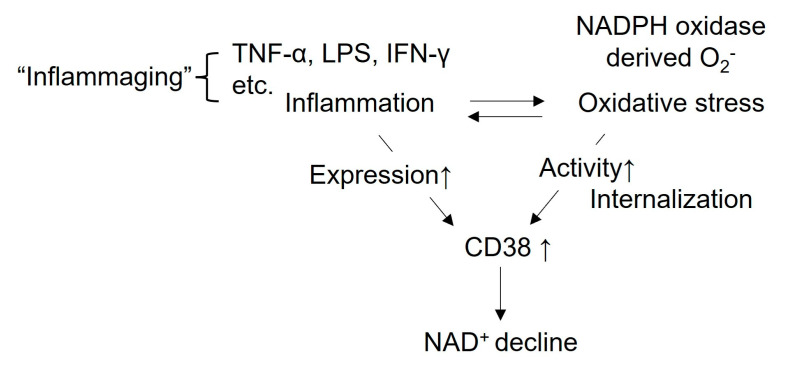
Regulation of CD38 by inflammation and oxidative stress. CD38 expression is enhanced by inflammatory cytokines including TNF-α, LPS and IFN-γ, etc., leading to cellular NAD^+^ decline. Since inflammation is closely linked to aging, the relationship between inflammation and aging is called “inflammaging”. In addition, oxidative stress derived from NADPH oxidase activates CD38 by its internalization in the plasma membrane. A decline in NAD^+^ decreases the activation of NAD^+^-dependent enzymes, sirtuins such as Sirt1 and Sirt3, resulting in metabolic dysfunction, promoting the aging process and the progression of age-related diseases through inflammation, oxidative stress, mitochondrial dysfunction and a reduction in autophagy.

**Figure 4 cells-12-00595-f004:**
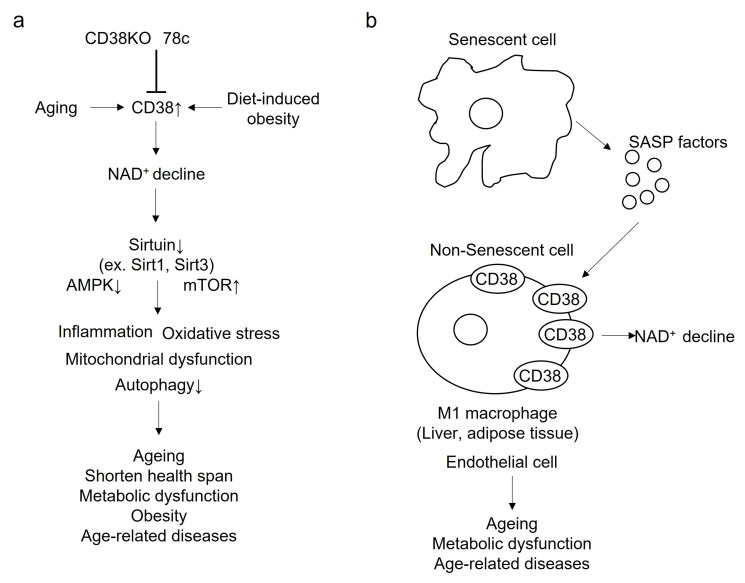
(**a**). Beneficial effect of gene ablation and pharmacological inhibition of CD38 on aging and age-related diseases. CD38 activity increases with aging or diet-induced obesity, leading to NAD^+^ decline. Thereafter, NAD^+^ decline contributes to the promotion of aging, a shortened health span, metabolic dysfunction, obesity and age-related diseases, through the alteration of cellular function including inflammation, oxidative stress, mitochondrial dysfunction and decreased autophagy. This is via a reduction in sirtuins such as Sirt1 and Sirt3. CD38 knockout mice and the CD38 specific pharmacological inhibitor, 78c, may reduce aging, lengthen health span, and reduce metabolic dysfunction, obesity and age-related diseases. (**b**). Relationship between CD38 activation by SASP factors derived from senescent cells in terms of age-related diseases. Although senescent cells do not have a high expression of CD38, the SASP factors including inflammatory cytokines secreted by senescent cells induced CD38 expression and its activation in non-senescent cells, such as proinflammatory M1-like macrophages. Accumulating a higher amount of CD38-expressed M1-like macrophages in tissues such as the white adipose tissue and liver during chronological aging causes NAD^+^ to decline in the tissues, further promoting the aging process and metabolic dysfunction. In addition, the increased expression of CD38 was observed in endothelial cells and macrophages in response to SASP factors, possibly leading to the initiation and progression of cardiovascular disease, which is related to aging.

**Table 1 cells-12-00595-t001:** Role of CD38 in heart disease.

The Protective Effect of CD38 Deficiency/Inhibition against Heart Injury			
Experimental Model	Animal/Cells	Effect	Reference
Ischemia/reperfusion Hypoxia/reoxygenation	CD38KO mice (C57BL/6 mice) CD38 knockdown H9c2 cells	Infarct size↓ Sirt1↑ FoxO1,3↑ SOD2↑ catalase↑ Nox4↓ Ca^2+^↓	[108]
Hypoxia/ischemic thermal burn model	CD38KO mice (B2.129P2-CD38tm1Lnd backcrossed to a C57BL/6J for more than 10 generations) Cultured neonatal mouse cardiomyocytes	PLEKHM1↑ Sirt1↑ FoxO1↑ Rab7↑ Autophagy↑	[109]
High-fat diet	CD38KO mice (C57BL/6 mice) CD38 knockdown H9c2 cells with oleic acid	GSH↓ FFA↓ NAD^+^↑ ROS↓lipid synthesis↓ Sirt3↑FoxO3↑SOD2↑ Nox2↓Nox4↓	[110]
Ang-II infusion	CD38KO mice (C57BL/6 mice) CD38 knockdown H9c2 cells	Cardiac hypertrophy↓ Sirt3↑AMPK↑ FoxO3↑ SOD2↑catalase↑ ERK↓Akt↓GSK3β↓ Ca^2+^↓NFATc↓	[113]
d-galactose (d-gal)-induced myocardial cell senescence	CD38 knockdown H9c2 cells CD38 knockdown with Sirt1 specific inhibitor Ex-527	SA-*β*-Galactose↓ ROS↓Nox4↓SOD2↑ Autophagy↑ Cellular senescence↑ ROS↑	[114]
Male mice	CD38KO male (C57BL/6 mice)	Serum testosterone↑ RyR2↑ SERCA2↑ NCX1↑α-MHC↑ Myocardial contraction↑	[115]
Caffeine–epinephrine stimulation	CD38KO mice (C57BL/6 mice) Genetically modified mice with catalytically inactive of CD38 WT mice treated with antibody 68	Exercise capacity↑ Basal heart rates↓ Heart rate variability↑ SERCA2a↑ Frequency of spontaneous Ca^2+^ release under stressful conditions↓ Ventricular tachycardia↓	[116]
Isoproterenol stimulation	CD38KO mice (C57BL/6 mice)	Arrhythmogenicity↓	[55]
Duchenne muscular dystrophy model, isoproterenol stimulation	mdx/CD38KO mice	Cardiac function↑ Cardiac fibrosis↓ Mortality due to heart failure↓ hypertrophy↓ Ca^2+^ sparks and waves in cardiomyocytes at rest↓	[117]

CD38KO, CD38 knockout; α-NAD, nicotinamide adenine dinucleotide; FoxO, forkhead box protein O; SOD2, super oxide dismutase 2; PLEKHM1, pleckstrin homology and run domain containing M1; glutathione, GSH; FFA, free fatty acid; ROS, reactive oxygen species; Ang-II, angiotensin-II; AMPK, AMP-activated kinase; ERK, extracellular-signal-regulated kinase; GSK-3β, glycogen synthase kinase 3β; NFATs, nuclear factor of activated T-cells; SA-*β*-Gal, senescence-associated beta-galactosidase; ryanodine receptor 2, RyR2; sarco-endoplasmic reticulum Ca^2+^ ATPase 2, SERCA2; sodium–calcium exchanger 1, NCX-1; α- myosin heavy chain, α-MCH; wild type, WT; vascular smooth muscles, VSMCs.

**Table 2 cells-12-00595-t002:** The role of CD38 on vascular tissue and cells.

CD38 Deficiency/Inhibition Protects Vascular Tissue And Cells			
Experimental Model	Animal/Vascular Tissue/Cells	Effect	Reference
Ang-II-induced hypertension	CD38KO mice (C57BL/6 mice) Cultured mice VSMCs	Hypertension↓ Vascular remodeling↓ VSMCs senescence↓ Sirt1↑Mitophagy↑ Sirt3↑SA-sEVs biosynthesis/secretion↓	[118]
Ischemia/reperfusion	CD38 knockdown or CD38 inhibitor α-NAD in cultured RAECs C57BL/6J mice treated with α-NAD	NADP(H)↑ NO↑O_2_^−^↓BH4↑ Recovery of contractile function↑ Infarct size of the LV↓	[119]
Hypoxia/reoxygenation	RAECs with CD38 knockdown	NADP(H)↑ NO↑O_2_^−^↓	[121]
Ischemia/reperfusion	CD38KO mice (C57BL/6 mice)	NADP(H)↑ NO↑O_2_^−^↓BH4↑GSH↑ Recovery of contractile function↑ Infarct size of the LV↓	[122]
Ischemia/reperfusion	Sprague Dawley rats treated with luteolinidin	NADP(H)↑ NO↑O_2_^−^↓BH4↑ Recovery of contractile function↑ Coronary flow↑ Infarct size of the LV↓	[120]
High-glucose condition	Mice VSMCs with CD38 knockdown	Inflammasome↓ VSMCs proliferation↓ Collagen I synthesis↓	[123]
**CD38 deficiency/inhibition promotes coronary atherosclerosis**			
High-fat diet	CD38KO mice (C57BL/6 mice) Coronary artery myocytes from CD38KO mice	Coronary artery collagen I deposition↑ Autophagy↓Collagen I degradation↓/accumulation↑	[124]
High-fat diet	CD38KO mice (C57BL/6 mice) coronary artery Coronary artery SMCs from CD38KO mice	Autophagy↓ Inflammasome↑ Inflammation↑ Remodeling↑	[125]
High-fat diet	CD38KO mice (C57BL/6 mice) coronary artery Bone-marrow-derived macrophages from CD38KO mice	Intimal and media layer thickening↑ Aggregation of macrophages↑ Deposition of free cholesterol↑ NAADP induced Ca^2+^↓ Free cholesterol efflux from lysosome↓	[126]
7-ketocholesterol stimulation	CAMs from CD38KO mice (C57BL/6 mice)	cADPR/NAADP induced Ca^2+^↓ O_2_^−^ via Nox4↓ Nrf2↓ SMCs proliferative and atherosclerotic phenotype↓	[127]

Ang-II, angiotensin-II; VSMCs, vascular smooth muscle cells; SA-sEVs, Senescence-associated-small extracellular vesicles; RAECs, rat aorta endothelial cells; NO, nitric oxide; BH4, tetrahydrobiopterin; LV, left ventricle; NAADP, nicotinic acid adenine dinucleotide phosphate; SMCs, smooth muscle cells; CAMs, coronary artery myocytes; cADPR, cyclic ADP-ribose; Nrf2, nuclear factor erythroid 2-related factor 2.

**Table 3 cells-12-00595-t003:** Role of CD38 on kidney disease.

The Protective Effect of CD38 Deficiency/Inhibition against Kidney Injury			
Experimental Model	Animal/Cells	Effect	Reference
Diabetic kidney disease High-glucose condition	Zucker diabetic fatty rats treated with apigenin HK-2 cells treated with apigenin	NAD^+^/NADH↑ Sirt3↑ Acetylated-SOD2↓Acetylated-IDH2↓ Mitochondrial ROS↓ Tubulo-interstitial inflammation↓fibrosis↓	[133]
LPS-induced AKI	C57BL/6 mice treated with quercetin CD38 knockdown quercetin RAW 264.7 cells	Tubular injury↓ Inflammation↓ Serum Urea↓ Macrophage M1 polarization↓ NF-κB↓inflammation↓	[74]
**CD38 deficiency or inhibition in promoting kidney injury**			
Doca+1% NaCl treatment	CD38KO mice (C57BL/6 mice) C57BL/6J mice with CD38 shRNA Conditionally immortalized mouse podocytes with CD38 shRNA	Podocytes EMT↑ Glomerular injury↑ Blood pressure↑ Urinary protein/albumin excretion↑	[134]
Cell culture	Conditionally immortalized mouse podocytes with CD38 shRNA	NAADP-mediated Ca^2+^ release↓ Lysosome fusion to autophagosome↓	[135]
LPS-induced AKI	CD38^−/−^ mice (B6.129P2-Ly96^−/−^)	Serum Cr↑ Tubular injury↑ Inflammation↑ NF-κB↑ TRL4↑IFN-*γ*↑	[138]

HK-2, human kidney 2; NADH, nicotinamide adenine dinucleotide; SOD2, superoxide dismutase 2; IDH2, isocitrate dehydrogenase 2; ROS, reactive oxygen species; CD38KO, CD38 knockout; Doca, deoxycorticosterone acetate; AKI, acute kidney injury; NF-κB, nuclear factor-kappa B; EMT, epithelial-mesenchymal transition; NAADP, nicotinic acid adenine dinucleotide phosphate; shRNA, small hairpin RNA; LPS, lipopolysaccharide; Cr, creatinine; TRL4, toll-like receptor 4; IFN-*γ,* interferon-*γ*.

## Data Availability

Not applicable.

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
