# Peer review of "The Role of CD38 in the Pathogenesis of Cardiorenal Metabolic Disease and Aging, an Approach from Basic Research"

_cells, 2023, doi:10.3390/cells12040595_

Round 1

Reviewer 1 Report

In the manuscript entitled “The role of CD38 in the pathogenesis of cardiorenal metabolic 2 disease and aging”,the authors reviewed literatures on NAD+ synthesis and degradation, enzymatic activity of CD38, and roles of CD38 on ageing/metabolic dysfunction/obesity/cardiovascular-kidney diseases? The review was well written. The phenotypic observations in this study are highly interesting and clinically relevant, and the mechanistic investigations are in-depth. The manuscript would be further improved by addressing the following concerns.

Major Concerns:

1. Most literatures reviewed in this manuscript were from mouse studies. Therefore, the title needs to be changed. Authors should also discuss the link between mouse results and human diseases. More importantly, authors should discuss whether CD38 is a therapeutic target in cardiorenal metabolic disease and aging.   

2. It is better to combine section 4 and section 5 since insulin secretion is a part of metabolic process.

3. Section 7 on implication of CD38 in COVID-19 infection pathophysiology should not be included in this review.

4. Literatures related to signaling pathways in CD38 regulation should be discussed.

5. Roles of CD38 overexpression in ROS production and protein degradation should be discussed, e.g., Stem Cells. 2015 Sep;33(9):2664-73; Cell Death Dis. 2022 Nov 9;13(11):944.

Minor points:  

Line 15: exist→exists

Line 63-64: 'numerous basic studies' needs to be specified.

Line 181: type III CD38 exists in the membranes of intracellular organelles and also on cell membranes.

Author Response

Response to Reviewer 1 Comments

We thank reviewers and editors for proving us another chance to revise our manuscript. Authors are providing answers to all the comments by point to point raised by reviewers’ comments.  

Major Concerns:

  1. Most literatures reviewed in this manuscript were from mouse studies. Therefore, the title needs to be changed. Authors should also discuss the link between mouse results and human diseases. More importantly, authors should discuss whether CD38 is a therapeutic target in cardiorenal metabolic disease and aging.

Response 1;

We thank reviewer’s constructive comments on our manuscript.
We changed the title to “The role of CD38 in the pathogenesis of cardiorenal metabolic disease and aging, an approach from basic research”.
We changed the section of “Conclusion” to “Conclusions and prospectives”, as below.

  1. Conclusions and prospectives

In this review, we summarized the role of CD38 in the pathogenesis of aging and cardio-renal-metabolic diseases, primarily from reports of basic research using animal models. NAD+ plays an important role in an array of cellular processes such as mitochondrial function and metabolism, oxidative stress, inflammation, cell signaling, cell division and DNA repair. Therefore, dysregulation of NAD+ metabolism is related to aging and age-related metabolic derangement and multiple diseases. CD38 is the main NADase in mammalian tissues, and the role of CD38-activation-related NAD+ decline has been investigated in relation to the pathogenesis of the aging process, metabolic dysfunction and age-related diseases. Numerous studies in basic research have shown that the inhibition of CD38 may be a potential therapeutic target for aging and cardiorenal metabolic disease.
    Conversely, several studies have reported that inhibition of CD38 may lead to the exacerbation the pathological conditions such as insulin secretion, and cardiovascular and kidney disease. This discrepancy may be based on the feature of CD38, functioning not only as the regulator of NAD+, but also the regulator of intracellular Ca2+ homeostasis via cADPR and NAADP. Therefore, the role of CD38 in the pathogenesis of several disease requires further evaluation, both in terms of NAD+ metabolism and intracellular Ca2+ homeostasis.
   Could CD38 be a potential therapeutic target for human aging and cardiorenal metabolic disease? In clinical practice, CD38-targeted therapy using CD38 monoclonal antibodies has been performed in the patients with hematologic malignancies such as multiple myeloma (MM), which highly expresses CD38 on their malignant cells [139,140,141,142,143]. CD38 is expressed not only on hematological malignant cells in MM, but also on some solid tumor cells. Interestingly, the role of CD38 in solid tumor cells has shown conflicting data, indicating that CD38 may be involved in both tumor progression and suppression depending on tumor type [144,145,146,147]. In addition, CD38 is expressed on tumor-related immunosuppressive cells, including regulatory T and regulatory B cells and myeloid-derived suppressor cells [139,148]. Therefore, treatment with CD38 antibodies may exhibit anti-tumor activity by releasing T-cell suppression via inhibition of such immunosuppressive cells, and may be the therapeutic potential for solid tumors beyond hematologic malignancies such as MM. Follow-up data regarding the cardiorenal outcomes or metabolic changes including diabetes on the patients treated with CD38 antibodies may be useful to clarify the role of CD38 on cardiorenal metabolic disease in humans. However, since there are no previous reports showing such clinical data, the role of CD38 in human aging and the pathophysiology of cardiorenal metabolic disease remains unclear.
    Thus, although structural variations of the molecule or specific association with different molecules according to the organ, tissue or cell may produce different functional effects respectively, CD38 has both physiological and pathological roles, and great potential as one of the candidate molecules for anti-aging. However, further studies are necessary to clarify whether the CD38 may be a therapeutic target for human aging and age-related diseases, including cardiorenal metabolic disease.

  1. It is better to combine section 4 and section 5 since insulin secretion is a part of metabolic process.

Response 2; We combined section 4 and section 5.

  1. Section 7 on implication of CD38 in COVID-19 infection pathophysiology should not be included in this review.

Response 3; We deleted the section 7 in this review.

  1. Literatures related to signaling pathways in CD38 regulation should be discussed.
    Response 4;

We thank reviewer’s constructive comments on our manuscript.
We added the discussion regarding signaling pathway in CD38 expression, as below.

“Tumor necrosis factor-α (TNF-α) activates the mitogen-activated protein kinases (MAPKs) such as the extracellular signal-regulated kinase (ERK), p38, and c-Jun N-terminal kinase (JNK). Although p38 and JNK upregulates CD38 expression through the activation of NF-κB and AP-1, ERK and p38 may be involved in stabilizing the CD38 transcripts [77,78].”

[77] Tirumurugaan KG, Jude JA, Kang BN, Panettieri RA, Walseth TF, Kannan MS. TNF-alpha   induced CD38 expression in human airway smooth muscle cells: role of MAP kinases and transcription factors NF-kappaB and AP-1. Am J Physiol Lung Cell Mol Physiol. 2007 Jun;292(6):L1385-95.
[78] Guedes AG, Deshpande DA, Dileepan M, Walseth TF, Panettieri RA Jr, Subramanian S, Kannan MS. CD38 and airway hyper-responsiveness: studies on human airway smooth muscle cells and mouse models. Can J Physiol Pharmacol. 2015 Feb;93(2):145-53.

5. Roles of CD38 overexpression in ROS production and protein degradation should be discussed, e.g., Stem Cells. 2015 Sep;33(9):2664-73; Cell Death Dis. 2022 Nov 9;13(11):944.
Response 5;
    On the other hand, other report has shown that high expression of CD38-induced NAD+ decline increases oxidative stress through downregulation of antioxidation-related genes [86]. Ferroptosis is recognized as a form of cell death driven by iron-dependent lipid peroxidation, and it may be involved in cellular aging [87,88]. Ma et al. demonstrated that in bone marrow derived macrophages from aged mice, increased expression of CD38 may trigger oxidative degradation of dihydrofolate reductase (DHFR) through sulfonation of the Cys 7 residue, increasing susceptibility of ferroptosis [89]. Thus, CD38 and oxidative stress interact and may be involved in cellular aging.

Ref.
[86] Hu Y, Wang H, Wang Q, Deng H. Overexpression of CD38 decreases cellular NAD levels   and alters the expression of proteins involved in energy metabolism and antioxidant defense. J Proteome Res. 2014;13:786–95.
[87] Stockwell BR. Ferroptosis turns 10: Emerging mechanisms, physiological functions, and therapeutic applications. Cell. 2022 Jul 7;185(14):2401-2421.
[88] Larrick JW, Larrick JW, Mendelsohn AR. Contribution of ferroptosis to aging and frailty. Rejuvenation Res. 2020;23:434–8.
[89] Ma Y, Yi M, Wang W, Liu X, Wang Q, Liu C, Chen Y, Deng H. Oxidative degradation of dihydrofolate reductase increases CD38-mediated ferroptosis susceptibility. Cell Death Dis. 2022 Nov 9;13(11):944. 

Wei and colleagues previously demonstrated that the CD38 signaling plays an important role in regulating cell proliferation and neuronal differentiation in PC12 cells (1). In addition, they showed that in in the studies using mouse embryonic stem cells (ESCs), CD38 signaling inhibits the cardiomyocyte differentiation (2), but is required for neural differentiation, through CD38 signaling-mediated ROS production (3). In this review, we mainly focus on the role of CD38 on aging and age-related diseases, therefore, we did not include the data regarding neuron and cell differentiation of ESCs this time.

Ref.
(1) Yue J, Wei W, Lam CM et al. The CD38/ cADPR/Ca2+-pathway promotes cell proliferation and delays NGF-induced differentiation in PC12 cells. J Biol Chem 2009;284:29335– 29342.
(2)Wei WJ, Sun HY, Ting KY et al. Inhibition of cardiomyocytes differentiation of mouse embryonic stem cells by CD38/cADPR/Ca2+ signaling pathway. J Biol Chem 2012;287: 35599–35611.
(3)Wei W, Lu Y, Hao B, Zhang K, Wang Q, Miller AL, Zhang LR, Zhang LH, Yue J.
CD38 Is Required for Neural Differentiation of Mouse Embryonic Stem Cells by Modulating Reactive Oxygen Species. Stem Cells. 2015 Sep;33(9):2664-73. 

Minor points: 
Line 15: exist→exists

Response; We corrected to “exists” on line 15.

Line 63-64: 'numerous basic studies' needs to be specified.

Response; We changed to “numerous basic studies as described below.” on line 63-64.

Line 181: type III CD38 exists in the membranes of intracellular organelles and also on cell membranes.

Response;
We changed to “In addition, type III CD38 exists in the membranes of intracellular organelles including the nucleus, mitochondria, endoplasmic reticulum, endolysosomes and lysosomes and also on cell membranes facing the cytosol, and plays an important role in regulating the level of intra-organelle NAD+, cADPR and Ca2+ [53,65,66,68,69].

Reviewer 2 Report

In this review, the Authors discussed the role of CD38 in aging and the pathogenesis of age-related diseases including cardiorenal metabolic diseases. The Authors highlight the two “faces” of CD38: NAD-consumer, and also a mediator in the production of Ca2+ mobilizing messengers.

The topic is interesting and deserve publication, but the Authors need to take a further effort in correcting it. The use of the English language must be improved, also in view of the fact that some sentences convey a message that is not the correct one. Sometimes this is due to the use of the language, and also sometimes I fear it is due to the fact that the Authors did not fully understood the article they are discussing.

Thus, I strongly encourage the Authors to go through their manuscript, making sure that every sentence and the message of every quoted study are clear.

Here I list some of the changes that are needed (I cannot list every improper use of the English language).

1.       Line 21: “is activated” (not “activates”)

2.      Line 93: remove “to”

3.      Please check that NAD is always written with + as superscript

4.      Line 118: “quinolinic” (not “quinolitic”)

5.      Line 155 and 503: “ADP-ribosyl cyclase” is the correct way to define the CD38-mediated enzymatic activity producing cADPR from NAD. cADP-risbosyl cyclase is wrong.

6.      Line 160. Just delete this sentence, which contradicts the previous ones.

7.      Line 167-169 (and also Fig 2B). The Authors should at least once also quote the studies according to which NAADP is not produced by CD38, and the studies according to which NAADP can activate ryanodine receptors (not on lysosome), and not only TPC.

8.      Line 174: rephrase. I suppose they meant “orientations” and not “proteins”

9.      Line 178: remove “transportation to inside”, remove “to”

1.  Line 184: the Authors should quote PMID: 15650244, regarding the role of CD38 type II in the production of paracrinally relevant cADPR.

1.  Table 1. I might have overlooked it, but in ref 95 I did not read the use of a-NAD. Please double-check this point and all the data summarized in the Tables.

1.  Line 443. Please rephrase (the decline in NAD+ contributes to both NADP+ and NADPH). Do the Authors mean “the decline in NAD+ contributes to both NADP+ and NADPH decline”?

1.  Line 505-506. As it is, this sentence is wrong. Please rephrase. “Both NAD+ and NADP+ are consumed by CD38, resulting in the generation of NADH and NADPH”. CD38 is not generating NADH and NADPH!

Author Response

Response to Reviewer 2 Comments

We thank reviewers and editors for proving us another chance to revise our manuscript. Authors are providing answers to all the comments by point to point raised by reviewers’ comments.  

  1. Line 21: “is activated” (not “activates”)

Response 1; We corrected to “is activated” on line 21.

  1. Line 93: remove “to”

Response 2; We removed “to” on line 93.

  1. Please check that NAD is always written with + as superscript

Response 3; We rechecked and corrected to NAD written with + as superscript.

  1. Line 118: “quinolinic” (not “quinolitic”)

Response 4; We corrected to “quinolitic” on line 118.

  1. Line 155 and 503: “ADP-ribosyl cyclase” is the correct way to define the CD38-mediated enzymatic activity producing cADPR from NAD. cADP-risbosyl cyclase is wrong.

Response 5; We corrected to “ADP-ribosyl cyclase” on line 155 and 503.

  1. Line 160. Just delete this sentence, which contradicts the previous ones.

Response 6; We deleted this sentence.

  1. Line 167-169 (and also Fig 2B). The Authors should at least once also quote the studies according to which NAADP is not produced by CD38, and the studies according to which NAADP can activate ryanodine receptors (not on lysosome), and not only TPC.

Response 7; NAADP elicits Ca2+ release from the two-pore channel (TPC) receptor located in acidic lysosomes or endolysosomes, or from the RyR on ER, having implications for intra-cellular Ca2+ homeostasis [58,59,60,61,62]. (Figure 2b). In addition to CD38, NAADP is generated by NADPH oxidase or dual NADPH oxidase (DUOX), from NAADPH. (Figure 2b) [63, 64].

Ref.

[58] Calcraft P.J., Ruas M., Pan Z., Cheng X., Arredouani A., Hao X., Tang J., Rietdorf K., Teboul L., Chuang K.T., et al. NAADP mobilizes calcium from acidic organelles through two-pore channels. Nature. 2009;459:596–600. doi: 10.1038/nature08030.

[59] Brailoiu E., Churamani D., Cai X., Schrlau M.G., Brailoiu G.C., Gao X., Hooper R., Boulware M.J., Dun N.J., Marchant J.S., et al. Essential requirement for two-pore channel 1 in NAADP-mediated calcium signaling. J. Cell Biol. 2009;186:201–209.

[60] Ruas M, Rietdorf K, Arredouani A, Davis LC, Lloyd-Evans E, Koegel H, Funnell TM, Morgan AJ, Ward JA, Watanabe K, Cheng X, Churchill GC, Zhu MX, Platt FM, Wessel GM, Parrington J, Galione A. Purified TPC isoforms form NAADP receptors with distinct roles for Ca(2+) signaling and endolysosomal trafficking.

Curr Biol. 2010 Apr 27;20(8):703-9.

[61] Hohenegger M., Suko J., Gscheidlinger R., Drobny H., Zidar A. Nicotinic acid-adenine dinucleotide phosphate activates the skeletal muscle ryanodine receptor. Biochem. J. 2002;367:423–431. doi: 10.1042/bj20020584. 

[62] Langhorst M.F., Schwarzmann N., Guse A.H. Ca2+ release via ryanodine receptors and Ca2+ entry: Major mechanisms in NAADP-mediated Ca2+ signaling in T-lymphocytes. Cell. Signal. 2004;16:1283–1289. doi: 10.1016/j.cellsig.2004.03.013.

[63] Ogunbayo, O.A.; Zhu, Y.; Rossi, D.; Sorrentino, V.; Ma, J.; Zhu, M.X.; Evans, A.M. Cyclic

adenosine diphosphate ribose activates ryanodine receptors, whereas NAADP activates two-

pore domain channels. J. Biol. Chem. 2011, 286, 9136-9140. doi: 10.1074/jbc.M110.202002

[64] Pitt, S.J.; Funnell, T.M.; Sitsapesan, M.; Venturi, E.; Rietdorf, K.; Ruas, M.; Ganesan,

A.; Gosain, R.; Churchill, G.C.; Zhu, M.X.; et al. TPC2 is a novel NAADP-sensitive Ca2+

release channel, operating as a dual sensor of luminal pH and Ca2+. J. Biol. Chem. 2010,

285, 35039-35046. doi: 10.1074/jbc.M110.156927

  1. Line 174: rephrase. I suppose they meant “orientations” and not “proteins”

Response 8;

CD38 situates in two opposite membrane orientations, with extracellular and cytosolic catalytic sites, as type II and III, respectively [65,66].

  1. Line 178: remove “transportation to inside”, remove “to”

Response 9; We removed “to” on line 178.

  1. Line 184: the Authors should quote PMID: 15650244, regarding the role of CD38 type II in the production of paracrinally relevant cADPR.

Response;

Further, type II CD38 paracrinally supplies a neighboring concentrative nucleoside transporter (CNT)-and RyR-expressing cells with cADPR, regulating intracellular Ca2+ mobilization and its related cellular functions, in several cells such as smooth myocytes, 3T3 murine fibroblasts, hippocampal neurons, and human hemopoietic stem cells [70, 71].

Ref.

[70] De Flora A, Zocchi E, Guida L, Franco L, Bruzzone S. Autocrine and paracrine calcium signaling by the CD38/NAD+/cyclic ADP-ribose system. Ann N Y Acad Sci. 2004 Dec;1028:176-91

[71] Astigiano C, Benzi A, Laugieri ME, Piacente F, Sturla L, Guida L, Bruzzone S, De Flora A. Paracrine ADP Ribosyl Cyclase-Mediated Regulation of Biological Processes. Cells. 2022 Aug 24;11(17):2637.

  1. Table 1. I might have overlooked it, but in ref 95 I did not read the use of a-NAD. Please double-check this point and all the data summarized in the Tables.

Response;

We re-checked all the data summarized in Table 1. In ref. 95 of Table 1, we corrected “CD38KO mice supplied with a-NAD” to “CD38KO mice”. And also, in the text, we corrected “CD38 deficiency suppled with a-NAD” to “CD38 deficiency”.

  1. Line 443. Please rephrase (the decline in NAD+ contributes to both NADP+ and NADPH). Do the Authors mean “the decline in NAD+ contributes to both NADP+ and NADPH decline”?

Response;

We deleted “the decline in NAD+ contributes to both NADP+ and NADPH” on line 443. And also, we changed “activation of CD38” to “activation of the NAD(P)ase function of CD38”, on line 445.

  1. Line 505-506. As it is, this sentence is wrong. Please rephrase. “Both NAD+ and NADP+ are consumed by CD38, resulting in the generation of NADH and NADPH”. CD38 is not generating NADH and NADPH!

Response;

We changed the sentence to “The NADP+ are consumed by CD38, consequently contributing to depletion of NADP(H) pool, which is the substrate of NADPH oxidase” on line 505-506.

Reviewer 3 Report

The manuscript by Dr. Koya is a complete and accurate review discussing the role of CD38 in aging and age-related diseases. The role of CD38 prevalently known as a surface molecule widely used in hematology is the focus of different interest and the groups with variable backgrounds. For these reasons, the authors start from a complete and clear definition of the potential roles played by the molecule in different contexts. The graphical part is complete and clear and helps understanding a complex biological network. Then the authors analyzed the positive and negative contribution of the CD38-ruled products to the genesis and definition of the disease under scrutiny, in different organs.

The interesting observation is that the Koya group summarized and expanded the observation by Chini et al about SASP factors in aging and age-related diseases. The conclusion of their work may be dual. One side the conclusion is that the inhibition of CD38 may acquire a therapeutic potential. On the other, the authors have the honesty to admit that the picture may be more articulated and CD38 inhibition may be followed by exacerbation of distinct pathologies. One should note that most of the experiments are done in mice: murine CD38 shows difference on the human counterpart.

The authors reach a conclusion derived from studying human CD38 with monoclonal antibodies. CD38 is a multipotent molecule. One of the aspects not quoted is represented by its being an adhesion molecule, with CD31 acting as a natural ligand. CD31 (also known as PECAM-1) is highly expressed by endothelial cells (J Immunol. 1998 Jan 1;160(1):395-402.PMID: 9551996 and J Immunol. 1996 Jan 15;156(2):727-34. PMID: 8543826). A second conclusion is related to the existence of specific antibodies not only blocking, while a limited set of these reagents is agonistic (J Immunol. 1990 Oct 15;145(8):2390-6. PMID: 1976692 and doi: 10.1182/blood. v99.7.2490). A reasonable conclusion is that structural variations of the molecule or peculiar association with different molecules according to the organ or tissue may give rise to different functional effects.

Lastly, we assist in medicine to the extensive use of human (or humanized) anti-CD38 antibodies for the treatment of multiple myeloma. Results indicate that the same antibody reacting with the same molecule is followed by cytotoxicity to the cell target, but simultaneously by activation of positive effects on normal killer T lymphocytes. At the same time, there is elimination or blocking of normal regulatory T lymphocytes (doi: 10.1182/blood-2017-06-740944).  Relevant is also a set of effects or signals implemented by ligation of the molecule by the therapeutic antibody as observed on myeloma cells (doi: 10.1111/bjh.17329).

In my view, for a follow-up paper the authors should investigate myeloma patient after antibody-therapy. Further, they may have access to patients treated with Daratumumab (no inhibition of the enzymatic activities) or with Isatuximab (reported as blocking the production of cADPR (doi: 10.3390/cells8121522).

Author Response

Response to Reviewer 3 Comments

We thank reviewers and editors for proving us another chance to revise our manuscript. Authors are providing answers to all the comments by point to point raised by reviewers’ comments.  

The manuscript by Dr. Koya is a complete and accurate review discussing the role of CD38 in aging and age-related diseases. The role of CD38 prevalently known as a surface molecule widely used in hematology is the focus of different interest and the groups with variable backgrounds. For these reasons, the authors start from a complete and clear definition of the potential roles played by the molecule in different contexts. The graphical part is complete and clear and helps understanding a complex biological network. Then the authors analyzed the positive and negative contribution of the CD38-ruled products to the genesis and definition of the disease under scrutiny, in different organs.

The interesting observation is that the Koya group summarized and expanded the observation by Chini et al about SASP factors in aging and age-related diseases. The conclusion of their work may be dual. One side the conclusion is that the inhibition of CD38 may acquire a therapeutic potential. On the other, the authors have the honesty to admit that the picture may be more articulated and CD38 inhibition may be followed by exacerbation of distinct pathologies. One should note that most of the experiments are done in mice: murine CD38 shows difference on the human counterpart.

The authors reach a conclusion derived from studying human CD38 with monoclonal antibodies. CD38 is a multipotent molecule. One of the aspects not quoted is represented by its being an adhesion molecule, with CD31 acting as a natural ligand. CD31 (also known as PECAM-1) is highly expressed by endothelial cells (J Immunol. 1998 Jan 1;160(1):395-402.PMID: 9551996 and J Immunol. 1996 Jan 15;156(2):727-34. PMID: 8543826). A second conclusion is related to the existence of specific antibodies not only blocking, while a limited set of these reagents is agonistic (J Immunol. 1990 Oct 15;145(8):2390-6. PMID: 1976692 and doi: 10.1182/blood. v99.7.2490). A reasonable conclusion is that structural variations of the molecule or peculiar association with different molecules according to the organ or tissue may give rise to different functional effects.

Lastly, we assist in medicine to the extensive use of human (or humanized) anti-CD38 antibodies for the treatment of multiple myeloma. Results indicate that the same antibody reacting with the same molecule is followed by cytotoxicity to the cell target, but simultaneously by activation of positive effects on normal killer T lymphocytes. At the same time, there is elimination or blocking of normal regulatory T lymphocytes (doi: 10.1182/blood-2017-06-740944).  Relevant is also a set of effects or signals implemented by ligation of the molecule by the therapeutic antibody as observed on myeloma cells (doi: 10.1111/bjh.17329).

In my view, for a follow-up paper the authors should investigate myeloma patient after antibody-therapy. Further, they may have access to patients treated with Daratumumab (no inhibition of the enzymatic activities) or with Isatuximab (reported as blocking the production of cADPR (doi: 10.3390/cells8121522).

Response ;

We changed the section of “Conclusion” to “Conclusions and prospectives”, as below.

  1. Conclusions and prospectives

In this review, we summarized the role of CD38 in the pathogenesis of aging and cardio-renal-metabolic diseases, primarily from reports of basic research using animal models. NAD+ plays an important role in an array of cellular processes such as mitochondrial function and metabolism, oxidative stress, inflammation, cell signaling, cell division and DNA repair. Therefore, dysregulation of NAD+ metabolism is related to aging and age-related metabolic derangement and multiple diseases. CD38 is the main NADase in mammalian tissues, and the role of CD38-activation-related NAD+ decline has been investigated in relation to the pathogenesis of the aging process, metabolic dysfunction and age-related diseases. Numerous studies in basic research have shown that the inhibition of CD38 may be a potential therapeutic target for aging and cardiorenal metabolic disease.
    Conversely, several studies have reported that inhibition of CD38 may lead to the exacerbation the pathological conditions such as insulin secretion, and cardiovascular and kidney disease. This discrepancy may be based on the feature of CD38, functioning not only as the regulator of NAD+, but also the regulator of intracellular Ca2+ homeostasis via cADPR and NAADP. Therefore, the role of CD38 in the pathogenesis of several disease requires further evaluation, both in terms of NAD+ metabolism and intracellular Ca2+ homeostasis.
   Could CD38 be a potential therapeutic target for human aging and cardiorenal metabolic disease? In clinical practice, CD38-targeted therapy using CD38 monoclonal antibodies has been performed in the patients with hematologic malignancies such as multiple myeloma (MM), which highly expresses CD38 on their malignant cells [139,140,141,142,143]. CD38 is expressed not only on hematological malignant cells in MM, but also on some solid tumor cells. Interestingly, the role of CD38 in solid tumor cells has shown conflicting data, indicating that CD38 may be involved in both tumor progression and suppression depending on tumor type [144,145,146,147]. In addition, CD38 is expressed on tumor-related immunosuppressive cells, including regulatory T and regulatory B cells and myeloid-derived suppressor cells [139,148]. Therefore, treatment with CD38 antibodies may exhibit anti-tumor activity by releasing T-cell suppression via inhibition of such immunosuppressive cells, and may be the therapeutic potential for solid tumors beyond hematologic malignancies such as MM. Follow-up data regarding the cardiorenal outcomes or metabolic changes including diabetes on the patients treated with CD38 antibodies may be useful to clarify the role of CD38 on cardiorenal metabolic disease in humans. However, since there are no previous reports showing such clinical data, the role of CD38 in human aging and the pathophysiology of cardiorenal metabolic disease remains unclear.  

Thus, although structural variations of the molecule or specific association with different molecules according to the organ, tissue or cell may produce different functional effects respectively, CD38 has both physiological and pathological roles, and great potential as one of the candidate molecules for anti-aging. However, further studies are necessary to clarify whether the CD38 may be a therapeutic target for human aging and age-related diseases, including cardiorenal metabolic disease.

Round 2

Reviewer 2 Report

Please correct. 

As I previously wrote: quinoliNic is the correct word.

Also, carefully read again the manuscript. There are still errors (s at the third person that are still missing, etc)